# Human Vault RNAs: Exploring Their Potential Role in Cellular Metabolism

**DOI:** 10.3390/ijms25074072

**Published:** 2024-04-06

**Authors:** Magdalena Taube, Natalia Lisiak, Ewa Totoń, Błażej Rubiś

**Affiliations:** Department of Clinical Chemistry and Molecular Diagnostics, Poznan University of Medical Sciences, Rokietnicka 3, 60-806 Poznan, Poland; magdalena.taube@ump.edu.pl (M.T.); nlisiak@ump.edu.pl (N.L.); etoton@ump.edu.pl (E.T.)

**Keywords:** vault RNA, drug resistance, apoptosis, autophagy, cancer, aging, cellular metabolism, non-coding RNAs, cancer therapy

## Abstract

Non-coding RNAs have been described as crucial regulators of gene expression and guards of cellular homeostasis. Some recent papers focused on vault RNAs, one of the classes of non-coding RNA, and their role in cell proliferation, tumorigenesis, apoptosis, cancer response to therapy, and autophagy, which makes them potential therapy targets in oncology. In the human genome, four vault RNA paralogues can be distinguished. They are associated with vault complexes, considered the largest ribonucleoprotein complexes. The protein part of these complexes consists of a major vault protein (MVP) and two minor vault proteins (vPARP and TEP1). The name of the complex, as well as vault RNA, comes from the hollow barrel-shaped structure that resembles a vault. Their sequence and structure are highly evolutionarily conserved and show many similarities in comparison with different species, but vault RNAs have various roles. Vaults were discovered in 1986, and their functions remained unclear for many years. Although not much is known about their contribution to cell metabolism, it has become clear that vault RNAs are involved in various processes and pathways associated with cancer progression and modulating cell functioning in normal and pathological stages. In this review, we discuss known functions of human vault RNAs in the context of cellular metabolism, emphasizing processes related to cancer and cancer therapy efficacy.

## 1. Introduction

The pool of non-coding RNAs consists of a wide spectrum of RNA species with a broad size range, from the very small endogenous non-coding microRNAs (17–25 nt) to large, long non-coding RNAs (more than 200 nt). Members of the non-coding RNA family have various cellular functions and are characterized by different biogenesis pathways. Depending on the subclass, they are transcribed by different RNA polymerases (rRNAs are Pol I transcripts, miRNAs and lncRNAs are typically transcribed by RNA Pol II, while vtRNAs are Pol III transcripts). Moreover, they also differ in post-transcriptional processing mechanisms and their ability to interact with proteins [1].

While quite a lot is known about microRNAs and their mode of action in the regulation of target gene expression via post-transcriptional mRNA cleavage, destabilization, or finally via inhibition of translation (reviewed in [2]), the role of vault RNAs in the regulation of gene expression or their biogenesis pathways, processing, or editing needs to be elucidated. Vault RNAs (vtRNAs) were described in the mid-1980s as a class of small non-coding RNAs, found together with MVP (major vault protein), TEP1 (telomerase-associated protein 1), vPARP (vault-associated poly(ADP-ribose) polymerase), in the largest identified so far eukaryotic ribonucleoprotein particles (RNPs), termed “vaults” [3]. Interestingly, vault complexes and vault RNAs were discovered during rat liver preparation and protein fractionation as small ovoid bodies with unknown structures and functions. Further processing, staining, and observations under the electron microscope revealed structures with complex morphology and shape resembling cathedral vaults, and that is where their name comes from [3]. Vaults are massive barrel-shaped particles with a size of 40 × 40 × 70 nm and a molecular weight of ~13 MDa [4,5,6]. These assemblies comprise 78 copies of major vault protein (MVP) and additional proteins, TEP1 and vPARP [7]. Vaults are mainly located in the cytoplasm, but some reports confirmed their localization in the nuclear membrane also, within or near the nuclear pore complex (NPC) [8,9,10]. The number of vaults is highly conserved and varies from 10,000 to 100,000/cell, depending on species [11,12]. The vault RNP structure is characterized by rapid redistribution in response to external stimuli or stresses and is hypothesized to facilitate protein exchange between the cytoplasm and nucleus. Furthermore, the vault complex and its components are considered mediators of nuclear–cytoplasmic translocation in normal and cancer cells because the shuttle of these molecules between cellular compartments in response to external factors involved in tumorigenesis may be crucial for cellular vault function [7,13,14,15]. Interestingly, confocal laser microscopy and cryo-immunoelectron microscopy assessment in the U373 glioblastoma cell line revealed nuclear localization of part of the MVP protein [16]. The role of TEP1, MVP, and vPARP is reported in various cellular processes and diseases, such as multidrug resistance, apoptosis, cancer progression, and epilepsy [17,18,19,20,21,22,23,24,25,26,27]. The protein components of vaults are also considered risk and/or prognostic factors in various cancer types [20,28,29,30]. Vault RNAs were initially thought to be solely located in vault particles, but literature data demonstrated that less than 5% are bound to vault complexes. The majority of vault RNAs (95%) are not associated with vault complexes and are located evenly in the cytoplasm as free, being unassociated with vault molecules [31,32,33]. Moreover, published data strongly support the hypothesis suggesting that vault complexes bind the RNA molecules with different affinities depending on the cellular metabolism status and other factors. The binding affinity and the ratio between free and bound RNAs to the vault complex might be functionally important [9]. The ratio between the level of RNA molecules inside and outside the vault complex may be potentially crucial for using vault RNAs as markers and/or targets in human diseases. However, since the association status of the vault RNAs with the vault complex affects and modifies their action, the exact role of vault RNAs still needs to be discovered. Regarding the role of vault RNA in various cellular processes and their involvement in multiple mechanisms, the first evidence came from studies on Epstein–Barr virus (EBV) infection in B cells. The studies showed that upregulation of vault RNA transcripts during infection reduced infected cell apoptosis (inhibition of the extrinsic and intrinsic apoptotic pathways). The first report showed that the effect is mediated not via MVP but via specific action of a vault RNA, namely, *vtRNA1-1*. Interestingly, the central domain of vault RNA that distinguishes vtRNA1-1 from its paralogues is essential for its function, i.e., apoptosis resistance [32,34,35]. Non-coding vault RNAs are found in many eukaryotes. In the human genome, four vault RNA paralogues (*vtRNA1-1*, *vtRNA1-2*, *vtRNA1-3*, and *vtRNA2-1*) are found. RNA polymerase III transcribes them, and they are ca. 100 nt long (*vtRNA1-1* 98 nt, *vtRNA1-2* 88 nt, *vtRNA1-3* 88 nt, *vtRNA2-1* 100 nt) [36]. In this review, we present the state of the art concerning vault RNAs and their role in cellular processes, involvement in human diseases, and therapy.

## 2. Vault RNA Genes in Humans

The vtRNA1-1 gene shows the highest expression level among all vault RNA-coding genes, potentially resulting from a unique secondary B-box (B2) element [9]. Regarding the structure of vault RNA-coding genes, it is worth noting that a polymerase III type 2 promoter controls all vault RNA genes and contains box A and box B motives, commonly present in tRNA-coding genes [37]. Interestingly, the box A and box B sequences in *VTRNA-1* and *VTRNA-2* are crucial and bind transcription factors TFIIIC and TFIIIB, which facilitate polymerase III binding to the transcription start site. Thus, these elements play a critical role in tight control of polymerase III activity and cellular homeostasis control [38]. However, functional vault RNA genes also have many upstream elements that may distinguish active vtRNA genes from untranscribed ones. Vault RNAs are rich in secondary structure elements (they form imperfect hairpin structures by base pairing; 2D structures of four human vault RNA paralogues are presented in the graphical abstract and Figure 1 in detail), and their role remains unclear [36,39]. Some single-stranded regions may pair with other transcripts, possibly with proteins and small molecules.

Interestingly, vault RNAs are similar to another enigmatic but better-described family of ncRNAs called Y-RNAs, another class of Pol III transcripts [36,43]. Y-RNAs are Ro60 ribonucleoprotein particles that function in immune cell communication, immunopathology, antiviral immunity, and HIV-1 infection pathways [44]. Y-RNAs, like other RNA Pol III transcripts and due to their secondary structure mimicking viral structure, may act as innate immune guardians during cell infection with viruses. Y-RNAs are cellular RIG-I ligands mobilized upon HIV-1 infection. Thus, they can trigger RIG-I-dependent immune response [44]. These molecules may be critical in maintaining cellular functions, because their high abundance in human plasma [45] makes them good biomarker candidates, e.g., in breast cancer (BC). Notably, BC patients have been characterized by increased levels of Y-RNA [45,46].

Although vault RNAs may play similar functions, they are less conserved, and it is suggested that they emerged recently in the evolutionary timeline [36]. Consequently, the number of vault RNA genes varies between vertebrates, and are absent in bacteria, fungi, nematodes, and plants. Both classes of RNAs have a similar size, are controlled by similar promoters, appear in a single or small number of functional copies, and are typically transcribed from a small gene cluster. Interestingly, both families (vault RNA and Y-RNA) are stably related to their adjacent protein-coding genes for unknown reasons, and it is hypothesized that this association may significantly impact their function [36,43]. The only significant distinction is the composition of the Y-RNA cluster in tetrapods (four or five ancient paralogues). In contrast, clustered vault RNA paralogues are evolutionarily younger and have arisen independently multiple times. Because the molecular function of vault RNAs remains unclear, speculation about the reason for lineage-specific evolutionary patterns seems pointless [36]. The entire, larger genome fragment was copied during evolution in various organisms [36,43,47].

In humans, four vault RNAs are encoded on chromosome 5q31 in two loci. The *VTRNA-1* locus contains the genetic information for three vault RNAs (*vtRNA1-1*, *vtRNA1-2*, and *vtRNA1-3*) and is located between the zinc-finger matrin-type 2 gene and proto-cadherin cluster. Interestingly, protein-coding genes in the *vtRNA1-2* neighborhood play similar roles. This suggests an association at the expression-modulation level between genes coding for vault RNAs and adjacent proteins. However, the molecular basis of this functional link requires further research [36,48]. The *VTRNA-2* locus encodes the information for *vtRNA2-1* and is located between the genes coding for transforming growth factor beta 1 and SMAD family member 5 [11,32,34,36,48].

## 3. Vault RNAs and Their Regulatory Role in Cancer

Cancer is a complex genetic disease arising from elaborate genome changes. These changes include a cumulative collection of gain-of-function mutations that induce oncogenes, loss-of-function mutations that inactivate tumor suppressor genes, and mutations that inactivate stability genes involved in proliferative cell division, all of which help facilitate the transformation of a normal cell to a malignant phenotype [49].

In general, non-coding RNAs are critical players in transcription regulation, and they can control the post-transcriptional regulation of mRNA under physiological conditions. The role and mode of action of microRNAs or lncRNAs to date have been widely examined under physiological and pathological conditions, and various reports suggest their role in splicing, RNA editing or mRNA degradation, protecting genomes from external nucleic acids, involvement in DNA synthesis or genome rearrangement [50]. Moreover, altered expression of non-coding RNAs can also contribute to several pathologies, including cancer [13], genetic disorders, including DiGeorge syndrome, phenylketonuria, and neurodevelopmental defects (comprehensively reviewed by Mattick et al. [51]). Notably, the unknown mechanism of vault RNAs makes them the most mysterious among the various non-coding RNAs. Interestingly, as mentioned above, although vault RNAs were first identified as a component of vaults, data from sedimentation experiments clearly showed that only a fraction of vault RNAs are incorporated into vaults. These results suggest roles in various interactions and processes unrelated to ribonucleoparticles that they can be part of [11,33,36]. In 2000 and 2011, Hanahan and Weinberg proposed six hallmarks of cancer, constituting a logical outline for better understanding the diversity and complexity of neoplastic diseases. Among these characteristics, we can distinguish the ability to avoid apoptosis, insensitivity to antigrowth signals, self-sufficiency in growth signaling, limitless replicative potential, sustained angiogenesis, and increased invasion and metastasis. It has been underlined that the process normal cells must undergo to achieve a neoplastic state is multistep, and a succession of these hallmarks is required. These changes include signaling aberrations during cell proliferation, the ability to evade growth suppressors, and constant replication. Moreover, cancer cells are characterized by resistance to death signaling and active, increased angiogenesis. These changes result in invasive and metastatic phenotypes [52,53]. Hanahan proposed a new, expanded set of functional capabilities acquired by human cells to develop from normal to neoplastic [54]. Accordingly, cancer cells are characterized by genome instability and mutations, deregulated cellular metabolism, the ability to evade growth suppressors, avoiding immune destruction, and sustaining proliferative signaling. Additionally, tumorigenesis is tightly associated with tumor-promoting inflammation and replicative immortality [55]. Like other non-coding RNAs, vault RNAs are key players in various cancer-related processes, i.e., gene expression, regulation of cell proliferation, apoptosis, autophagy, and drug resistance, which is the leading cause of chemotherapy failure [35,56,57,58].

## 4. vtRNA1-1

Although vault complexes and vault RNAs were discovered and described over 30 years ago, their exact role in processes during the cell life cycle in physiological and pathological conditions remains elusive and still needs further investigation.

To date, among the four mentioned vault RNA paralogues in humans, *vtRNA1-1* is the best known (Figure 2, Table 1) [6,55,56,59,60,61,62].

Many findings are compatible with the hypothesis that altered vault RNA expression is associated with tumorigenesis [6,31,57]. The novel prosurvival function of *vtRNA1-1* has been described, and these findings support the hypothesis that this molecule plays a crucial role in tumor cell proliferation, tumorigenesis, and chemoresistance (Figure 2). Cell proliferation analyses revealed that the knockout of *vtRNA1-1* but not *vtRNA1-3* in HeLa cells led to decreased cell proliferation [35]. A similar negative impact of *vtRNA1-1* knockout on cell proliferation was observed in the Huh-7 cell line (hepatocellular adenocarcinoma cells) [57].

Autophagy is a fundamental eukaryotic pathway that helps maintain metabolic homeostasis. This process is observed under physiological conditions, i.e., autophagy promotes embryonic organogenesis, such as neural tube formation, neuronal differentiation, reticulocyte development, myocardial growth, chondrocyte survival, and differentiation [69]. Autophagy is also observed in pathological conditions, i.e., inflammation, neurodegeneration, and cardiovascular diseases or carcinogenesis [70]. Autophagy in cancer is highly correlated with the innate and adaptive immune system and is mediated by cytokine signaling [71]. One of the critical autophagy markers, p62 protein, also called sequestosome 1 (SQSTM1), acts as a critical receptor that identifies and binds ubiquitinated proteins, delivering them via membrane-bound LC3 protein to the phagophore for subsequent degradation. p62 is also an RNA-binding protein that can engage small non-coding *vtRNA1-1* as a major interacting RNA. *vtRNA1-1* regulates p62-dependent autophagy via interference with p62 multimerization, a critical step for the biological activity of the p62 protein. *vtRNA1-1* directly modulates p62 oligomerization, which is critical for autophagy control. Specifically, *vtRNA1-1* acts as a riboregulator of key cellular processes via direct interaction/direct binding to the autophagic receptor p62, thus changing the protein function [72]. Horos et al. showed that overexpression of human *vtRNA1-1* inhibits, and its antisense LNA-mediated knockdown enhances p62-dependent autophagy [60]. Interestingly, starvation significantly reduces the steady-state and p62-bound levels of *vt RNA1-1*, which leads to autophagy induction [60].

These findings are also significant in tumorigenesis, as p62/SQSTM1 is the best-studied mammalian substrate thus far that shows autophagy specificity. Moreover, the modulation of this multidomain protein during autophagy is an inestimable factor in tumorigenesis. Data from extensive research employing knockout, transgenic, and knock-in mice strongly suggest its critical role in several cellular functions, including bone remodeling, obesity, and cancer development and progression (reviewed in [73]). To date, the mechanism through which *vtRNA1-1* can inhibit p62 oligomerization and the specificity of their interaction is lacking. Recently, Büscher et al. showed crucial components for binding between those two molecules [55]. It was found that the adjacent linker region of p62, PB1 domain, and central flexible loop within *vtRNA1-1* are necessary for specific binding with *vtRNA1-1*. Moreover, they revealed lysine 7 and arginine 21 as the two critical regulatory points of p62 [55].

Interesting results in the context of the regulatory role of *vtRNA1-1* in lysosome metabolism were published by Ferro and collaborators. Specifically, it was revealed that lack of *vtRNA1-1* in an in vitro and in vivo mouse model reduced drug lysosomotropism and inhibited tumor cell proliferation in hepatocellular carcinoma cells [57]. Moreover, vtRNA1-1 knockout in human HCC cells leads to lysosomal compartment dysfunction via TFEB nuclear translocation inhibition, resulting in a downregulation of the coordinated lysosomal expression and regulation (CLEAR) gene network. This network comprises several genes associated with lysosomal biogenesis, lysosomal acidification, and the autophagy pathway [74]. Thus, master gene transcription factor EB (TFEB) affects the expression of genes involved in lysosomal biogenesis, lysosome-to-nucleus signaling, and lipid catabolism. *vtRNA1-1* depletion resulted in increased activation of the mitogen-activated protein kinase (MAPK) cascade MAPK1/ERK2–MAPK3/ERK1, responsible for TFEB inactivation and cytoplasmic retention [57]. Kinases involved in signaling axes ERK1/2 MAPK and PI3K/Akt are described as autophagy regulators [75] and cell growth and proliferation modulators and are involved in tumorigenesis (reviewed in [76]). As one of the most critical cell cycle guards, the phosphoinositide 3 kinase (PI3K)/Akt signaling pathway is a major signaling pathway in various types of cancer. It controls hallmarks of cancer and is involved in angiogenesis and inflammation [77,78]. Thus, as mentioned above, signaling pathways are crucial for cancer development and progression and the patient therapy outcome.

Apoptosis is a tightly regulated and highly conserved type of cell death. This cascade of reactions and processes plays crucial roles in normal physiological conditions and pathological states, e.g., cancer. It involves the activation of caspases [79] but can also be associated with the MVP that was postulated as an apoptosis inhibitor in senescent cells [27]. However, very little is known about the role of vault RNA in apoptosis. In 2015, the first data concerning the general apoptotic resistance upon *vtRNA1-1* expression in malignant B cells was published. Amort and collaborators identified latent membrane protein 1 (LMP1), expressed in most EBV-associated lymphoproliferative diseases and malignancies [80], as a trigger for NF-κB-dependent *vtRNA1-1* expression. Ectopic expression of vtRNA1-1 in a B-cell line that usually lacks this ncRNA led to increased susceptibility to EBV infection due to attenuated apoptosis. Furthermore, knockdown experiments of the MVP clearly showed that the antiapoptotic effect is driven by *vtRNA1-1* and is not associated with the vault complex. Moreover, *vtRNA1-1* modulates the intrinsic and extrinsic apoptosis pathways [58]. By next-generation deep sequencing of the mRNome, Bracher and collaborators identified the PI3K/Akt pathway and the ERK1/2 MAPK cascade, which were dysregulated in the absence of *vtRNA1-1* during starvation-mediated cell death. They showed that the expression of 24 nucleotides of the vtRNA1-1 central domain seems crucial for maintaining apoptosis resistance. Interestingly, similar resistance to apoptosis in the lack of *vtRNA1-1* was also observed in two other cell lines (non-small-cell lung cancer A549 and human embryonic kidney HEK293 cells) [35]. These results strongly suggest that apoptosis is dependent on *vtRNA1-1* level.

Identifying efficient cancer treatment remains an important goal for both researchers and clinicians. Currently, chemotherapy is seen as one of the most promising cancer treatments to reduce the burden of cancer. However, chemotherapy fails in nearly 90% of cases, as cancer cells develop resistance to the anticancer agents, resulting in increased cancer invasion, progression, and metastasis [81]. Vault particles’ participation in drug resistance development in cancer cells has already been demonstrated. Moreover, MVP (also known as LRP, lung resistance-related protein) is generally overexpressed in human cancer cells and is involved in resistance phenotype formation [22,24,82]. According to Human Protein Atlas data, MVP is mainly accumulated in colorectal, renal, pancreatic, lung, and thyroid cancer (https://www.proteinatlas.org/ENSG00000013364-MVP; accessed on 27 February 2024). The role of vaults in therapy resistance may be associated with their ability to transport drugs away from their intracellular targets, e.g., from the nucleus or mediate the function of efflux pumps or exocytotic vesicles. Moreover, based on the characteristics of the minor vault proteins, vaults or vault components are likely involved in the maintenance of genomic stability (reviewed in [22]). Specifically, vault RNAs, especially *vtRNA1-1* and *vtRNA1-2*, were reported to bind mitoxantrone, one of the chemotherapeutic agents commonly used in malignancies, leading to the sequestration of this drug and consequently decreased treatment efficacy. Further investigation revealed overexpression of vault RNAs in cell lines derived from human glioblastoma, leukemia, and osteocarcinoma. Moreover, these cells showed increased resistance to mitoxantrone and vault RNA downregulation significantly decreased cell tolerance to this chemotherapeutic agent. These data strongly suggest an essential role of vault RNAs in susceptibility to mitoxantrone in tested malignant cells [59]. Although scientific data indicate the role of vault RNAs in small-molecule drug binding and mediating drug response, the mechanism still needs to be elucidated, as RNA molecules are rather unusual mediators for this type of drug.

Vault RNAs can mediate drug resistance in an MVP–vault complex-independent manner. By *vtRNA1-1* or MVP knockdown in MCF-7 cells, Chen et al. revealed that chemoresistance mediated by vtRNA did not depend on MVP and confirmed the role of *vtRNA1-1* in regulating PSF (polypyrimidine tract-binding protein-associated splicing factor) transcriptional activity independently of the expression of MVP [56]. It has been found that vtRNA1-1 can bind to the RNA-binding domain (RBD) on PSF protein, and when PSF expression was compromised, the cancer cells became sensitive to doxorubicin. On the other hand, the binding of *vtRNA1-1* to PSF released its target gene, GAGE6, and thus affected its expression level, and cells became more sensitive to the treatment. Thus, *vtRNA1-1* plays an important role in cancer cell drug resistance [56]. Moreover, it is suggested that a crucial function in drug resistance and cancer progression is played by lysosomes (reviewed in [83]).

Interestingly, literature data clarified the involvement of vault RNAs in acquiring an aggressive profile of TNBC (triple-negative breast cancer). It was shown that TNBC-associated *vtRNA1-1* was upregulated in estrogen receptor (ER)- and progesterone receptor (PR)-negative tumors, whereas *vtRNA1-2* seems to be associated only with ER-negative breast cancer. Furthermore, it is suggested that *vtRNA1-1* may discriminate cancer from normal tissue, inhibit apoptosis, and affect chemoresistance. These findings support the hypothesis suggesting the prognostic and therapeutic potential of vault RNAs in breast cancer, especially *vtRNA1-1* [63].

Consequently, it was shown as an excellent therapeutic biomarker for endocrine tumors [64]. Interestingly, the correlation between *vtRNA1-1* level and prognosis and/or response to treatment in endocrine tumors may be independent of tumor type and applied therapy but dependent on specific responsiveness. It was demonstrated that intracellular upregulation of vault RNAs and their specific exosomal release into the tumor microenvironment (TME) was directly correlated with successful tumor tissue damage after ASA 404 (vadimezan), EDPM (etoposide, doxorubicin, cisplatin, and mitotane) or LEDPM (etoposide, liposomal doxorubicin, liposomal cisplatin, and mitotane) administration [64]. Recently published data from Kato et al. revealed that in patients with blood diseases, serum *vtRNA1-1* levels might show some clinical significance [62]. Serum *vtRNA1-1* levels correlated with leukocyte counts and significantly increased in leukemia and lymphoma patients. During intensive chemotherapy treatment, the level of *vtRNA1-1* in sera decreased within these groups of patients. Notably, serum *vtRNA1-1* levels varied significantly in patients with hematological malignancies. Thus, the level of *vtRNA1-1* might be potentially used as a biomarker of normal and malignant hematological disorders [65]. Furthermore, this molecule may also be considered an index of therapy efficacy.

## 5. vtRNA1-2

*vtRNA1-2* cellular function and molecular mode of action remain very elusive and need further investigation. Recent studies published by Alagia et al. [48] revealed that Dicer-dependent small vault RNA1-2 (svtRNA1-2) processed/excised from the full-length *vtRNA1-2* is associated with Argonaute 2 (Ago2) and, in contrast to *vtRNA1-2*, is localized predominantly in the nucleus. It is postulated that *svtRNA1-2* can modulate the expression of genes coding for cell membrane proteins, such as proteins involved in signaling and glycoproteins. Furthermore, the knockdown of *vtRNA1-2* results in impaired cellular proliferation and upregulation of genes involved in cellular signaling and is associated with affected proliferation, cell adhesion, and migration (Table 2). Thus, vtRNA may act as a precursor for miRNA-like *svtRNA1-2*, but the detailed mechanism of its action is unknown.

## 6. vtRNA1-3

The role of *vtRNA1-3* in multidrug resistance was evaluated by van Zon and collaborators [9]. It was shown that elevated levels of *vtRNA1-3* may be mediated by a change in affinity between the vault complex and particular vault RNA in response to specific functional cues. Moreover, multidrug-resistant (MDR) cells were characterized by higher levels of *vtRNA1-3* associated with vault complexes. Consequently, the ratio of the vault-associated RNA species was suggested to influence the drug response [9]. Another interesting case showing the involvement of vault RNAs in tumorigenesis and patients’ prognosis was low-risk myelodysplastic syndrome (MDS). MDSs are heterogeneous clonal diseases characterized by ineffective hematopoiesis associated with increased apoptosis and differentiation block of early progenitors. It was shown that vault RNAs might also act as tumor suppressors in a specific manner based on epigenetic modifications, mainly DNA methylation. The hypothesis put forward by Helbo and colleagues [84] suggests that *vtRNA1-3* is silenced by DNA methylation in acute promyelocytic leukemia cells (HL60) but is unmethylated in normal hematopoietic cells (Table 3). Thus, the downregulation mentioned above seems to be cancer specific. Furthermore, cohort studies revealed that *vtRNA1-3* promoter hypermethylation was frequent in lower-risk MDS patients and was associated with decreased overall survival. However, analysis of the relative impact of *vtRNA1-3* promoter methylation on 5-year overall survival strongly suggests that it should not be considered an independent prognostic marker [84].

## 7. vtRNA2-1

Multiple studies have pointed to the possible role of *vtRNA2-1* in many different cancers (Table 4), interestingly acting as a tumor suppressor or oncogene depending on the cancer type. Those include the most relevant ones, like prostate and ovarian cancer. There exist two hypotheses concerning the potential mode of action of *vtRNA2-1*, an ncRNA whose significant role is inferred in the function of a partner protein, and another one concerning the transcript of *vtRNA2-1* as a non-canonical precursor of miRNA-like small RNAs (*snc886-3p* and *snc886-5p*). Research has unveiled significant differences in *vtRNA2-1* expression levels in prostate cancer tissue compared to normal prostate tissue [85,86].

The methylation status of the *vtRNA2-1* promoter increases from normal to tumor samples, whereas in the case of *vtRNA1-2*, the trend is the opposite.

Interestingly, low DNA methylation at the promoter region of *vtRNA1-1*, *vtRNA1-2*, and *vtRNA2-1* was associated with a lower patient survival rate [104]. Initially, the transcript of *vtRNA2-1* was annotated as a miRNA precursor after identifying small RNAs derived from its 3′ and 5′ UTRs. However, its role as a canonical miRNA precursor is questionable after identifying the direct inhibition of PKR (discussed later) by the full-length *vtRNA2-1* transcript. Nevertheless, the level of *snc886-3p* was altered in prostate cancer cell lines compared with regular prostate cell lines [86]. Fort et al. showed that *snc886-3p* is processed by DICER independently on DROSHA and associated with Argonaute 2 at a similar level as other microRNAs. This indicates that *snc886-3p* is functional because only small RNAs incorporated into the RISC complex are considered functional [86]. Supporting cell line experimental results, it was found that loss of *snc886-3p* expression was associated with early biochemical relapse in prostate tumors in patients [87]. Those findings support the role of *vtRNA2-1* as a tumor suppressor in prostate cancer. On the other hand, a higher plasma level of *snc886-3p* was associated with a higher grade of prostate cancer [105].

In the male population, prostate cancer shows the highest incidence and mortality among all cancer types and is characterized by high heterogeneity. Researchers found that *vtRNA2-1* dysregulation might be an essential and valuable prognostic marker in this type of cancer. It was found that upregulation of this ncRNA is not only associated with poor survival rate in patients but also significantly promoted proliferation and invasion, as this molecule was a tumor promoter. Moreover, the knockdown of *vtRNA2-1/miR-886* is related to inhibiting prostate cancer cell proliferation and invasiveness. Therefore, it can be considered an efficient therapeutic target [97]. *vtRNA2-1* is also perceived as a tumor suppressor gene (due to its ability to inhibit PKR) in the prostate [85,106], cholangiocarcinoma [88], skin [94], and gastric cancer cells [107]. Kunkaew and collaborators showed the importance of vtRNA2-1 suppression in cholangiocarcinoma [88]. They found that some cancer was characterized by elevated PKR activity in cholangiocyte cells [88]. This finding was surprising, because PKR is mainly known for its role as a proapoptotic factor through its ability to phosphorylate eIF2α in the host defense mechanism during viral infection. Researchers observed that *vtRNA2-1* was capable of suppressing PKR via direct physical interaction. The starting point for further research suggests a potential role of *vtRNA2-1* suppression and PKR activation during pre-malignant cell elimination in tumorigenesis. Thus, this is the evidence for PKR’s dual role during tumorigenesis. On the one hand, when the level of *vtRNA2-1* is low (or repressed), cells are directed toward canonically programmed cell death pathways. Interestingly, PKR plays a dual role in tumorigenesis. On the one hand, active PKR in cholangiocarcinoma cells does not induce apoptosis or an eIF2α-dependent pathway. On the other hand, it promotes the prosurvival NF-κB pathway [88].

Various evidence has revealed that vault RNA transcription is controlled by promoter gene methylation. Analysis of methylation status (and consequently expression profiling prediction) of the non-coding RNA promoters might be crucial in the context of the prognosis for cancer patients. In the case of hepatocellular carcinoma, a significant correlation between the methylation status of the *vtRNA2-1* promoter and various features of the tumor that affects the prognosis for the patient, such as the presence of large tumors, pathological vascular invasion, more advanced tumor stage or tumor recurrence, was identified. Moreover, the methylation of the *vtRNA2-1* promoter increased in stage III HCC when compared with less advanced stages. The methylation status of the *vtRNA2-1* promoter is also very beneficial for patient outcomes after liver resection. As suggested, patients with differentially hypermethylated *vtRNA2-1* promoters in tumor tissue compared to their adjacent normal tissue should receive more intensive care or adjuvant therapy after surgery [89].

Dugué and collaborators revealed that the methylation status of the *vtRNA2-1* region is heritable, modifiable, and highly important, being associated with increased cancer risk and a worse prognosis [90]. Interestingly, the results of the present cohort study showed no link between epigenetic factors and the individual’s lifestyle, i.e., age, smoking, diet, and blood cell DNA methylation at *vtRNA2-1*. These results might prove the leading role of genetics and adult lifestyle in explaining methylation variability at the heritable *vtRNA2-1* cluster. Moreover, investigated genetic and non-genetic factors are minimal in explaining variation in blood cell DNA methylation at *vtRNA2-1*. Thus, their potential role in the observed association between *vtRNA2-1* methylation status and breast cancer risk seems insufficient. Furthermore, the mechanism of DNA methylation inheritance in this region needs further investigation [90].

Genome methylation, one of the most critical mechanisms that control gene expression, also refers to genes coding for vault RNAs. As shown in acute myeloid leukemia (AML), *vtRNA2-1* can play a potential role in the progression and prognosis of AML. It was found that *vtRNA2-1* was monoallelically methylated in 75% of the control population, and the remaining 25% showed biallelic hypomethylation. It is worth noting that leukemia patients with unmethylated vtRNA2-1 showed better overall survival compared to monoallelic or biallelic methylation carriers. Moreover, *vtRNA2-1* induction (after demethylation provoked by 5-azanucleosides) was associated with a decreased level of phosphorylated PKR. Vault RNA2-1 modulates this kinase activity, which suggests that *vtRNA2-1* could be considered a tumor suppressor gene (TSG) in AML, since pPKR is known to act as a leukemic cell survival promoter. Thus, the methylation status of *vt RNA2-1* is a potential predictor of therapy outcome in AML patients [91].

Vault RNAs can directly interact with other non-coding RNAs and affect cancer cell proliferation. Recently published studies revealed the crucial role of direct interaction of *vtRNA2-1/nc886* with a novel lncRNA, i.e., *OXCT1-AS1*, in osteosarcoma progression by promoting tumor growth. Interestingly, this interaction is crucial for *vtRNA2-1* maturation suppression in this type of cancer. As reported, it plays an inhibitory role in cell proliferation and was demonstrated to contribute to cancer initiation, progression, and consequently metastasis [92,93]. It was also postulated to be associated with tumor evolution and cancer therapy response. Although the new signaling axis *OXCT1-AS1*–*vtRNA2-1* was identified, the mechanism and detailed functions at the molecular level of the downstream target genes of this axis remain unclear. This newly reported relationship requires further studies, not only in osteosarcoma but also in other cancer types, especially since the obtained results strongly suggest that overexpression of *OXCT1-AS1* and the ratio of full-length *vtRNA2-1* and *vtRNA2-1/nc886* after processing by Dicer might be useful prognostic parameters in osteosarcoma patients [92,93]. Moreover, *OXCT1-AS1* is known as an indicator of poor prognosis in glioblastoma (involved in the regulation of the *miR-195*–CDC25A axis [108]), as well as a metastasis promoter in non-small-cell lung cancer (by stabilizing LEF1/lymphoid enhancer factor 1) [95].

According to the data published by Hu et al. [96], higher levels of *vtRNA2-1* are expressed in the late phase of human endometrial cancer tissue, less than in the early phase, but still higher than in normal human endometrial tissue. Significantly, after *vtRNA2-1* silencing, the expression and protein levels of pPKR (phosphorylated PKR) and caspase 3 were increased, whereas NF-κB and VEGF were decreased. Furthermore, the apoptosis rate in the *vtRNA2-1*-silenced group was increased, and the cell proliferation rate was slower compared to the control [96]. Interestingly, opposite results indicating the oncogenic and antiapoptotic mode of action of *vtRNA2-1* were found in various cancer tissue types, i.e., endometrial [96], renal [109], as well as ovarian [110].

The role of vtRNAs was also reported in cervical squamous cell carcinoma (CSCC), the second-most common cancer in women worldwide [111]. Specifically, oncogenic activity and contribution to cervical cancer progression of *vtRNA2-1-5p* were reported. This non-coding RNA is AGO-associated small RNA derived from *vtRNA2-1* and it was upregulated in human cervical squamous cell carcinoma compared to normal tissue [98]. Interestingly, published data demonstrate that *vtRNA2-1-5p* has a specific role only in cervical cancers. Moreover, *vtRNA2-1-5p* can directly target p53 expression and act as an oncomir. As observed, its inhibition was associated with decreased proliferation and tumorigenicity and increased apoptosis of CSCC. Thus, *vtRNA2-1-5p* was suggested to be a direct regulator of p53 expression, playing an essential role in the apoptosis and proliferation of tumor cells. Furthermore, the inhibition of *vtRNA2-1-5p* sensitized cells to cisplatin treatment and directed them to the apoptotic pathway [98].

The literature data indicate that the expression of *vtRNA2-1-3p* can be specifically reduced, e.g., by ARE (arecoline, a nicotinic acid-based mild parasympathomimetic stimulant alkaloid found in the areca nut), which was reported in oral squamous cell carcinoma (OSCC). What is more, this suppression can play a critical role in the proliferation and migration of cancer cells and might affect the invasiveness of OSCC cells. These findings may be crucial in identifying new targets and treatment strategies in patients with this type of cancer [112].

Kunkaew and collaborators [113] discovered a cytotoxic mechanism based on the role of cellular non-coding RNA *vtRNA2-1* and PKR that acts as a proapoptotic protein. In normal cells, vtRNA2-1, right after its transcription, can bind to PKR and prevent PKR from aberrant activation. In cancer, it was observed that *vtRNA2-1* transcription was turned off after DOX treatment due to Pol III detachment from DNA. When the available pool of *vtRNA2-1* runs out, PKR does not have a binding partner, and cells are directed to apoptosis.

Furthermore, *vtRNA2-1* was identified as a molecular signal to sense DOX, and this finding explains why drugs that can cause DNA damage might also be cytotoxic for cells with the competent *vtRNA2-1*/PKR pathway [113]. One of the leading hypotheses on how vtRNAs and vault particles could be associated with drug resistance is based on PKR activity. Interestingly, it was shown that one of the vault RNA derivatives (*svRNAb*) is associated with Argonaute proteins and can drive specific cleavage, thus acting and regulating gene expression similarly to miRNAs. *svRNAb* can downregulate one of the most essential enzymes in drug metabolism—CYP3A4. This finding might be beneficial in the association of vault RNA and drug resistance. It can help explain the broad spectrum of potential mechanisms involved in these processes [62].

Cancer progression is associated with gene regulatory components such as transforming growth factor β (TGF-β) signaling and microRNAs. Higher tumor grades are characterized by elevated TGF-β signaling with simultaneous microRNA expression suppression. In patients with ovarian cancer, a high expression level of *vtRNA2-1/nc886* was associated with poor prognosis. In ovarian cancer, nc886 expression was induced by TGF-β, and *vtRNA2-1/nc886* inhibited microRNA maturation via binding to Dicer. Thus, *vtRNA2-1/nc886* regulates gene expression and processes such as cell adhesion, migration, and drug resistance [110]. This study explains potentially contradictory results concerning the role of *vtRNA2-1/nc-886* as a tumor suppressor or oncogene. Overall, one must be cautious when making conclusions from the in vitro studies, considering that small RNAs derived from full-length transcripts could have different roles than the precursor.

## 8. Vault RNAs in Aging

Undoubtedly, the occurrence and development of many diseases, including cancer, are related to aging. In recent years, an increasing number of researchers have concluded that immunological factors play an increasingly important role in physical degeneration and pathological changes, which may be essential targets for assessing and treating older individuals with cancer. Aging is a complex process that profoundly affects the immune system. The impairment of the immune system with age is manifested by increased susceptibility to infectious diseases, poorer response to vaccinations, autoimmune and other chronic diseases, and increased cancer incidence [114]. Recent reports indicate that overexpression of *vtRNA1-1* may attenuate apoptosis in a cellular model of Epstein–Barr virus infection [32,58] and accelerate apoptosis in influenza virus via PKR pathway (the major component of the cellular antiviral system) inactivation [115,116]. Furthermore, influenza A virus induced robust expression of vault RNAs in hosts through NS1 protein. In turn, the increased expression of vault RNAs was essential for NS1-mediated inhibition of PKR activation. These findings strongly support the hypothesis that viruses, including influenza A, can overcome PKR-mediated innate immune response through NS1-dependent upregulation of vault RNAs [116]. Since vault RNAs play an essential role in the epigenetic regulation of target gene expression, they are suggested to contribute to cancer or other disease development and aging. Kim and collaborators were interested in determining whether chemical compounds found in green tea that can control *vtRNA2-1* expression also have a role in the senescence of fibroblasts. Specifically, they tested AbsoluTea Concentrate 2.0 (ATC, absolute extract from green tea leaves obtained from *Camellia sinensis* var. sinensis cv. Jangwon No.3). They revealed that tested extracts led to increased *vtRNA2-1* expression that resulted in a suppression of cellular senescence that was manifested by a decrease in accumulation of cell senescence biomarkers, such as SA-β-gal activity and p16INK4A and p21Waf1/Cip1 expression. Thus, this non-coding RNA might be a good candidate for an antiaging target [99].

Similar observations were made by Lee and colleagues, who suggested the role of *vtRNA2-1* in controlling photoaging and inflammation in skin cells [94] (Figure 3). This study investigated how binding between vtRNA2-1 and PKR and regulation of PKR activity may affect skin aging. They showed that right after UVB radiation, the expression of *vtRNA2-1* decreases, whereas the phosphorylation status of PKR via MAPK kinases was increased. Consequently, together with reduced expression of *vtRNA2-1*, uncontrolled PKR activity, and increased expression of inflammatory cytokines (such as IL-1α, IL-6, and IL-8), cyclooxygenase (COX-2), collagenase type IV, MMP-9 (matrix metalloproteinase-9) (Figure 3), the inflammatory response and aging processes in the skin cells were significantly accelerated [94].

Eunsun Jung Lab went further with studies regarding the role of *vtRNA2-1* in photoaging and consistently obtained results suggesting that ultraviolet irradiation can accelerate the methylation of the *vtRNA2-1* gene [100]. Therefore, its expression is reduced, the PKR pathway is activated, and various proinflammatory factors are secreted. However, when *vtRNA2-1* was overexpressed, expression and production of those proinflammatory factors were inhibited. Moreover, the extract from *Laminaria japonica* was protecting *vtRNA2-1* against UVB, and the biosynthesis of UV-induced inflammatory factors mediated by the PKR pathway was inhibited (Figure 3) [100].

Moreover, overexpression of *vtRNA2-1* reduced the production of MMP-9 and inflammatory cytokines. In turn, UVB led to the inhibition of *vtRNA2-1* expression by increasing its methylation status. Simultaneously, UVB irradiation accelerated the phosphorylation of PKR, p38, JNK, c-Jun, and ATF-2. These results revealed that increasing *vtRNA2-1* expression, which is inhibited by UVB exposure, can inhibit the MMP-9 and inflammatory cytokines. The extract from *Laminaria japonica* suppressed PKR phosphorylation. Furthermore, treatment with *Laminaria japonica* extract inhibited the reduction of *vtRNA2-1* caused by UVB irradiation and blocked signal transduction via the PKR pathway to decrease the production of MMP-9 and inflammatory cytokines [100].

## 9. Vault RNAs and Immune Response

Literature data strongly suggest that *vtRNA2-1* suppresses IFN-β signaling and controls inflammation processes. In response to treatment with various pathogens, *vtRNA2-1* not only suppresses the activation of IRF3 (interferon regulatory factor 3) but also, via inhibition of PKR, can inhibit NF-κB (nuclear factor kappa-light-chain enhancer of activated B cells) and AP-1 (activator protein 1). These findings prove that *vtRNA2-1* is tightly connected with the regulation of expression of IFN-β and the factors needed to activate its promoter. Thus, *vtRNA2-1* is responsible for decreased expression of IFN-β and genes induced by interferon [101].

Interferons, innate immune response elements, mediate this reaction. Among various interferons, the first line of response to the pathogen activity is occupied by IFN-β. The impaired control of IFN-β might cause autoimmune diseases. Thus, the induction of this molecule and its secretion should be tightly controlled on various levels. *vtRNA2-1* might be a direct molecular link between the dependency of innate immunity on epigenetics and environmental factors such as UVB (Figure 3).

The development and proper functioning of the human immune system are closely related to and dependent on commensal microbiota. The microbiota plays emerging roles in the host’s innate and adaptive immune systems. The symbiosis between host and microbes is crucial for avoiding many immune-related disorders, such as inflammatory bowel disease (IBS), cardiometabolic disease, and cancer (reviewed in [117]). A recently published paper also showed the role of *vtRNA2-1* in disrupting the gut epithelial barrier via interaction with HuR, one of the RNA-binding proteins [102]. Increased levels of this vault RNA were observed in intestinal mucosal tissue samples from patients with IBS and samples from mice with colitis or sepsis. Higher ectopic expression of *vtRNA2-1* was associated with a decreased level of intracellular junction proteins, i.e., claudin 1, occludin, and e-cadherin, which resulted in intestinal epithelial barrier dysfunction in vitro. In contrast, *vtRNA2-1* silencing supports the functioning of the intestinal barrier. The results observed in mice agreed with those from in vitro experiments. Mechanistically, *vtRNA2-1* interaction with HuR prevents HuR binding to claudin 1 and occludin mRNA and decreases their translation [102].

## 10. Vault RNAs’ Role in Neural System Development and Pathology

*vtRNA1-1* functions as a riboregulator of synaptogenesis via MAPK signaling pathway modulation in neurons. In an in vitro synapse formation model, it was shown that murine vault RNA promotes synapse formation. Murine vault RNA is firstly transported to the distal part of neuronal cells as a component of the vault complex and subsequently released from the complex in the neurite by a mitotic kinase aurora A-dependent phosphorylation of MVP. Interestingly, the role in binding and activating MEK1 (mitogen-activated protein kinase) and enhancing MEK1-mediated ERK (extracellular signal-regulated kinases) activation in neurites by vault RNA per se, not as part of a ribonucleoprotein complex, suggests the existence of a novel molecular mechanism for synaptogenesis, with vault RNA a riboregulator in this process [66,67]. Further investigation revealed that only one human vault RNA paralogue could affect synaptogenesis. Expression of *vtRNA1-1* increases phosphorylation of ERK as well as the number of PSD95 (postsynaptic density protein 95) and synapsin I—proteins crucial for synapse maturation and biological function [68]. Interestingly, there is an emerging hypothesis associating vault RNA with an increased prevalence of ASD (autism spectrum disorder). The work of Geoffray and colleagues supports these claims. They found a link between dysregulation of the RAS MAPK signaling pathway and ASD. As MAPK is regulated by *vtRNA1-1*, it is postulated that vault RNA may also be associated with RASopathies—a group of inherited disorders caused by pathogenic variants of genes encoding regulatory proteins within the RAS-MAPK signaling pathway [118]. These findings need further investigation to find a direct link between vault RNA expression and ASD. Moreover, vault RNA expression in ASD could support the possible clinical translation of ASD-related MEK inhibitor drug therapy [119]. To date, it has been shown that ASD is not the only neurological pathology with the potential role of vault RNA in etiology as well as a potential therapy target. Miñones-Moyano and collaborators suggest that *vtRNA2-1* and its deregulation could contribute to pathogenesis in the initial stages of Parkinson’s disease (PD). It was observed that the expression level of this vault RNA was increased in the advanced stages of PD. However, the upregulation level was not strictly correlated with the severity of the neuropathological lesions. Researchers focused their studies on the amygdala in patients without motor symptoms in the early stages without pharmacological treatment. Thus, it might be excluded that drug treatment in PD patients could somehow affect the vault RNAs and other genes [103].

## 11. Conclusions and Future Perspectives

Vault RNAs play a role in cellular metabolism, and it is crucial to gain a better understanding of how these molecules are involved in normal cell metabolism, processes involved in tumorigenesis, and tumor response to treatment. Although vault complexes and vault RNAs have been known about for over 30 years, there is still an important gap regarding their mechanism of action and the molecular basis of their regulatory role. Vault RNAs and the whole ribonucleoprotein complexes that they are part of affect such important processes for cell functioning as autophagy, apoptosis and proliferation. Based on literature data, we can assume that vault RNAs may play a role in cell signaling control (e.g., PKR). Vault RNAs are also involved in neural cell development, synaptogenesis, metabolism and pathologies in the neural system, and neurodegenerative diseases, i.e., Parkinson’s disease or Alzheimer’s disease. This might be extremely important in the context of aging societies and the need to search for new mechanisms aimed at preventing, delaying or even repairing neuronal damage. Moreover, vault RNAs can contribute to multidrug response, especially *vtRNA1-1* and *vtRNA1-2*, via their ability to bind chemical compounds, including chemotherapy agents. Furthermore, since they can bind chemotherapeutic drugs, there is a hypothesis that they can also bind other chemicals, including xenobiotics. That fact might be crucial, due to growing environmental pollution levels and the need to counteract and prevent their effects on cellular metabolism. Vault RNAs and their regulatory role in a variety of pathways indicate that they may be considered as potential prognostic and diagnostic markers in cancer. As a subclass of pivotal regulators, vault RNAs, similarly to other non-coding RNAs, may be also crucial for cell development and differentiation, and together with their interaction network, may be involved in aging and aging-related disorders. Many non-coding RNA have been reported as key regulators and they can be involved in the regulation of cancer-related gene expression. It is extremely encouraging that also in the case of vault RNAs, it is possible to combine their activity and expression level with traditional therapy approaches for better prognosis and outcomes in cancer patients.

## Figures and Tables

**Figure 1 ijms-25-04072-f001:**
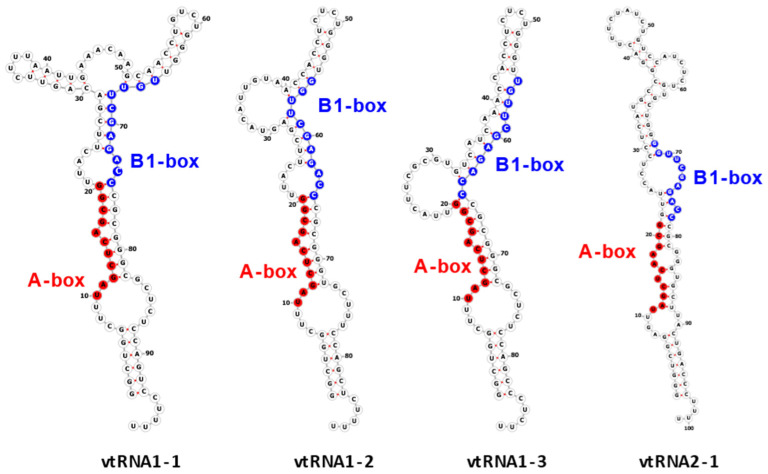
Secondary structure of human vault RNAs folded using RNAFold [40] and visualized using Forna [41]. A-box and B1-box Pol III internal promoter annotations were taken from [42].

**Figure 2 ijms-25-04072-f002:**
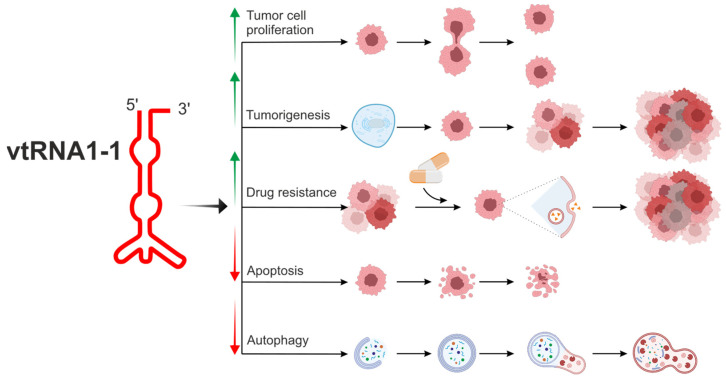
The biological potential of *vtRNA1-1* in the modulation of cell metabolism related to tumorigenesis and drug resistance. *vtRNA1-1* is the best known among four vault RNA paralogues in humans, and the *vtRNA1-1* gene shows the highest expression level in the group of vault RNA-coding genes. *vtRNA1-1* is involved in the regulation of crucial cellular processes associated with homeostasis. It can regulate apoptosis and autophagy, processes necessary to maintain the correct number of properly functioning cells. Moreover, this vault RNA is tightly associated with tumorigenesis, cell proliferation, and tumor ability to metastasize. *vtRNA1-1* is also considered a potential target in cancer therapy since its role in cell proliferation and drug resistance has been demonstrated. *vtRNA1-1* showed a prosurvival mode of action, stimulating tumor cell proliferation. Overexpression of *vtRNA1-1* led to autophagy, apoptosis inhibition, and drug resistance induction. Arrows indicate induction (↑) or inhibition (↓) of a specific mechanism/pathway.

**Figure 3 ijms-25-04072-f003:**
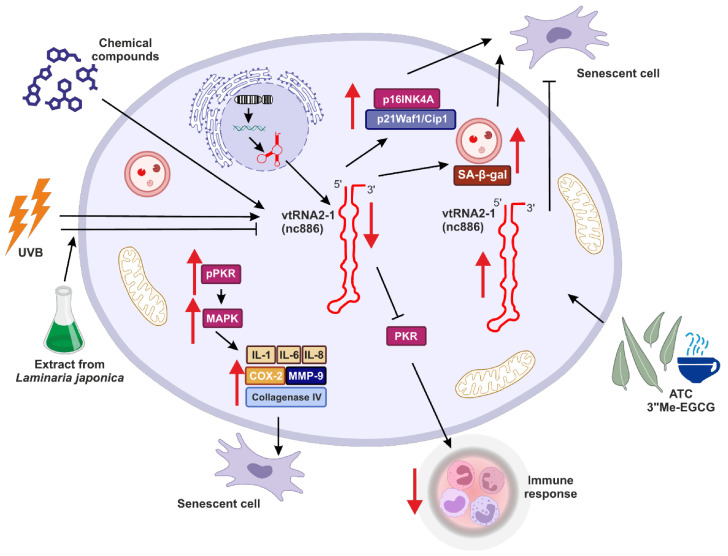
The potential role of *vtRNA2-1* in aging, senescence and immune response. *vtRNA2-1* expression may be controlled via various genetic and epigenetic mechanisms. This tight control is crucial for the role of *vtRNA2-1* in aging, and this molecule is a potential antiaging target. Red arrows indicate up- (↑) or downregulation (↓) of target effect.

**Table 1 ijms-25-04072-t001:** The role of human *vtRNA1-1* paralogue in cellular processes.

Structure	Expression	Pathophysiological Condition	Cells	Effect	Years	Ref.
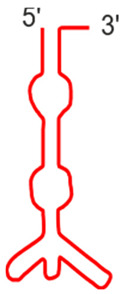	Up	EBV virus	----	Inhibition of apoptotic pathways	2015	[58]
	Down	----	HCC cells	Lysosomal compartment dysfunction via inhibited TFEB and downregulated CLEAR	2022	[57]
	Down		HeLa cells HuH-7 cells	Decreased cell proliferation	2020 2022	[35,57]
	Up	ER(-) and PR(-) breast cancer	----	----	2022	[63]
	Up	----	----	Inhibition of apoptosis, chemoresistance, cancer from normal tissue discrimination	2022	[63]
	----	Endocrine tumors	----	Prognostic marker, marker for therapy efficacy	2023	[64]
	Up	Leukemia, lymphoma	----	----	2023	[65]
	Up	----	HuH-7 cells	Inhibition of p62-dependent autophagy	2019 2022	[55,60]
	Down	----	HCC cells	Reduction in drug lysosomotropism, inhibition of cell proliferation, increased activation of MAPK cascade	2022	[57]
	Up	----	MG63, U118MG, U937, and U2OS cells	Mitoxantrone binding, chemoresistance induction	2005 2010	[6,59]
	Up	----	----	Induction of synaptogenesis via MAPK signaling pathways modulation in neurons	2021	[66,67,68]
	Up	----	----	Increasing of phosphorylation status of ERK and accumulation of PSD95 (postsynaptic density protein 95) and synapsin I, crucial proteins for synapse maturation and biological function	2021	[68]

EBV, Epstein–Barr virus; HCC, hepatocellular carcinoma; HuH-7, permanent cell line established from male hepatoma tissue; MG63, human fibroblast cell line; U118MG, human glioma cell line; U937, human myeloid leukemia cell line; U2OS, sarcoma cell line. ----: not applicable.

**Table 2 ijms-25-04072-t002:** The role of human *vtRNA1-2* paralogue in cellular processes.

Structure	Expression	Pathophysiological Condition	Cells	Effect	Years	Ref.
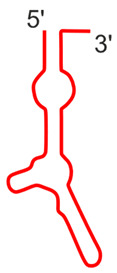	Down	----	----	Impaired cell proliferation and upregulation of genes associated with proliferation, migration, adhesion (svtRNA1-2)	2023	[48]
	Up	ER(-) breast cancer	----	----	2023	[48]
	Up	----	MG63, U118MG, U937, and U2OS cells	Mitoxantrone binding, chemoresistance induction	2005 2010	[6,59]
	Up	----	MCF-7	Mediated chemoresistance via PSF regulation	2018	[56]

MCF-7, breast cancer cell line; MG63, human fibroblast cell line; U118MG, human glioma cell line; U937, human myeloid leukemia cell line; U2OS, sarcoma cell line; PSF, polypyrimidine tract-binding protein-associated splicing factor.

**Table 3 ijms-25-04072-t003:** The role of human *vtRNA1-3* paralogue in cellular processes.

Structure	Expression	Pathophysiological Condition	Cells	Effect	Years	Ref.
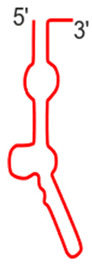	Down	Lower-risk myelodysplastic syndrome (MDS)	----	Silencing by DNA methylation in acute promyelocytic leukemia cells, unmethylated in normal promyelocytic cells	2015	[84]

**Table 4 ijms-25-04072-t004:** The role of human *vtRNA2-1* paralogue in cellular processes.

Structure	Expression	Pathophysiological Condition	Cells	Effect	Years	Ref.
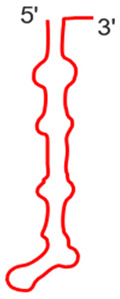	Down	Prostate cancer	DU145, PCSC1, PCSC2,PCSC3, RWPE, RWPE2, VCAP, LNCaP, WPE INT, PC3	Reduces cell cycle progression and increases apop-tosis (DU145, LNCaP, PC3)	2020	[63]
	Up (snc886-3p)	High grade Prostate cancer	----	----	2019	[64]
	Up (full-length nc886) Down (snc886-3p)	Osteosarcoma	Saos2, MG63	Promoting tumor growth (in cell ines)	2022	[87]
	Down	acute myeloid leukemia prostate cancer cholangiocarcinoma skin cancer gastric hepatocellular carcinoma stage III	----	----	2012 2020, 2018 2013 2019 2014 2020	[88] [89,90] [91] [92] [93] [94]
	Up	Endometrial Renal Ovarian	HEC-1A A-498	Oncogenic and anti-apoptotic	2017 2017 2018	[95] [96] [84]
	Down		PC3, LNCaP, DU145 and VCaP	Inhibition of prostate cancer cells proliferation and their invasiveness	2022	[97]
	Up	Cervical squamous cell carcinoma	HeLa, CSCC	p53 targeting, acting as an oncomir	2015	[98]
	Up	----	HDFs	Suppression of cellular senescence	2021	[99]
	Up	----	HaCaT	Controls photoaging and inflammation	2019	[94]
	Up	----	HaCaT	Reduces production of MMP-9 and inflammatory cytokines	2020	[100]
	Up	----	HEp-2, 293T, HCT116, Huh7, RAW264.7	Decreases expression of interferon β itself and genes induced by IFN- β	2021	[101]
	Down	----	Caco-2	Supports the functioning of the intestinal barrier	2023	[102]
	Up	Parkinson’s disease	----	Expression level was increased in advanced stages of PD	2013	[103]

DU145, human prostate cancer cell line; PCSC1, human prostate cancer cell line; PCSC2, human prostate cancer cell line; PCSC3, human prostate cancer cell line; RWPE, human prostate cancer cell line; RWPE2, human prostate cancer cell line; VCAP, human prostate cancer cell line; LNCaP, human prostate cancer cell line; WPE INT, human prostate cancer cell line; PC3, human prostate cancer cell line; Saos2, human osteosarcoma; MG63, human fibroblasts; HEC-1A, endometrial cancer cell line; A498, kidney cancer cell line; HDFs, human dermal fibroblasts; HaCaT, human keratinocyte; HEp-2, laryngeal cancer cell line; 293T, human embryonic kidney cell line; HCT116, human colorectal carcinoma cell line; RAW264.7, macrophage cell line; Huh7, human hepatoma cell line; Caco-2, colon carcinoma cell line; PD, Parkinson’s disease.

## Data Availability

No new data were created or analyzed in this study. Data sharing is not applicable to this article.

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
