# Peer review of "Human Vault RNAs: Exploring Their Potential Role in Cellular Metabolism"

_ijms, 2024, doi:10.3390/ijms25074072_

Round 1

Reviewer 1 Report

Comments and Suggestions for Authors

It is a very comprehensive review about human Vault RNAs. This review mainly summarized the diverse functions of human Vault RNAs. The contents are well welcome to deepen the knowledge of potential value of utilizing human Vault RNAs as a novel non-coding RNA therapy. Although this reveiw will stimulate the thoughtful discussion about the physiological function of Vault RNAs, it is still confusing about what kind of Vault RNAs can be used as the smart design of regulation approaches for cancer gene therapy. For the convenience of the readers, I suggest the author further refine this review according to the following points.

1) The title is confusing. There is no any discussion about the small molecules.

2) The fundamental mechanism of human Vault RNAs, especially in the aspect of target recognition is not clearly clucidated, although they have more than one target.

3) In each aspect of function, a potential mode of action of Vault RNAs should be provided. 

4) Since the target genes of Vault RNAs are numerous, the authors should provide a table to inform the promoter type, RNA length, target gene name, cancer cell type, etc.

5) The biogenesis and processing as mature Vault RNAs should be separately described. It is a very important aspect of regulating the abundance of Vault RNAs. For example, what kinds of nuclease process these Vault RNAs? What kinds of enzymes modify Vault RNAs? what kinds of nuclease degrade these Vault RNAs?

6) The potential application of Vault RNA therapy should be separately discussed.

7) The conclusion and perspective part is too long. A concise summary is highly recommended.

8) Some paragraphs are too long and should be shorten for the readability.

Author Response

First, we thank the reviewers for their kind comments and constructive criticism. Please find all comments addressed below and in the text. All suggested corrections were addressed, and the text was amended. All changes in the manuscript were made using a red font. Please find the response to the reviewers' comments point by point below.

Reviewer 2 Report

Comments and Suggestions for Authors

Author Response

(The authors gave the same response as above.)

Reviewer 3 Report

Comments and Suggestions for Authors

In the present manuscript, the authors explore the role of human Vault RNAs in Cellular Metabolism. In the manuscript, several Major Issues should be addressed in order to improve value of this paper.

1. First of all, the title does not appear to be clear and impactful. Authors should consider editing it. I would suggest a title like this below “Human Vault RNAs: Exploring their potential role in cellular metabolism” (in short, an incisive title).

2. In the “1. Introduction”: Authors should

-Check the wording of concepts in English. Many sentences are unclear due to a lack of correct punctuation and a clear explanation.

-Authors should consider discussing in this paragraph the:

1.  The pool of non-coding RNAs (I also suffer from quoting the following paper Orlandella FM, DOI: 10.1016/j.critrevonc.2022.103844)

2.      Discovery and history human vault RNA (vtRNA)

3. To make the topic more understandable, authors should consider creating

A.    A paragraph “2. human vault RNAs (vtRNAs)” in which it is defined in a general way:

• Biogenesis

• Localization

• Physiological function

• Structure (complex)

B.    A paragraph “3. vtRNA1-1” which discusses its role in

· proliferation

· autophagy

· drug resistance

· apoptosis

· aging

· immune system

associating it with the various pathologies discussed (i.e. cancer and neurological pathologies).

C.    A paragraph “4. vtRNA1-2”, “5. vtRNA1-3”, 6. vtRNA2-1”, inserting all the information contained in paragraphs 2. Non-Coding RNAs and Their Regulatory Role in Cancer, 3. The Role of vtRNAs in Cancer Cells Proliferation and Tumorigenesis, 4. vaultRNAs as Powerful Diagnostic and Cancer Therapy Monitoring Markers, 5. Epigenetic Regulation of vtRNAs Expression in Cancer, 6. Vault RNAs as a Regulator during the Autophagy, 7. Vault RNAs Act as Apoptosis Modulators, 8. vtRNAs and Drug Resistance, 9. Vault RNAs in Aging, 10. Vault RNAs and Immune Response, 11. vtRNAs Role in Neural System Development and Pathology. Paragraph 4, 5, 6 should therefore be discussed as paragraph 3, defined in point B.

4. Considering what is written in the previous points, we advise the authors to create a table for each vtRNA discussed (therefore dividing Table 1 into several tables).

Structure

Expression

Pathophysiological

condition

Cells

Effect

Years

Ref.

Up

EBV virus

…..

Inhibition of

apoptotic pathways

-----

[n°reference]

….

HCC cells

….

…..

Abbreviation: EBV, ……; HCC,…..; ns, not specified (where it is not defined in the cited paper).

Minor Issues:

-Check the spaces between words and references

-Check the punctuation

-Check the abbreviations

Author Response

(The authors gave the same response as above.)

Reviewer 4 Report

Comments and Suggestions for Authors

The review “Human Vault RNAs – Small Molecules and Great Perspectives. The Insight on Their Role in Cellular Metabolism” is devoted to vault RNAs and their role in cell processes and involvement in diseases. The vault RNAs is an interesting and prominent class of non-coding RNAs and the manuscript is properly written and comprehensive. Therefore, I recommend the review for publication in the International Journal of Molecular Science after minor revision.

I just have some remarks.

- line 75: the vtRNA paralogs are given in italics, whereas further they are don’t. Please, unify.

- line 162:  In his latest work, Hanahan did not propose new hallmarks of cancer, but only expanded them by offering four additional properties. It is possible to cite only the latest work of Hanahan 2022, since the general properties are the same.  

- The section 2 “Non-Coding RNAs and Their Regulatory Role in Cancer” is mostly devoted to the ncRNAs in general, but not vtRNAs. It could be reduced and combined with next sections. However, if the main point was to present the role of vtRNAs in each of mentioned hallmark of cancer, the next section should be numbered as subsections of the section 2.

- line 196: “ In the context of the vault RNAs role in cell proliferation and its regulation, another human paralogue was also analyzed.” Which paralogue? Please, specify.

- line 235: The “powerful diagnostic and cancer therapy monitoring markers” in the title of the section seems to be too strong for the “these molecule may be considered also as therapy efficacy index” in the end of the section. Please, replace “powerful” with another word.

- line 312 “It is concluded that PKR plays a dual role during tumorigenesis.” and line 315 “Interestingly, PKR plays a dual role in tumorigenesis.”. The same statement is repeated twice. Please, leave one sentence.

- line 318-353: These data don’t relate to the "epigenetic regulation of expression", they can be moved to the section 3 “The Role of vtRNAs in Cancer Cells Proliferation and Tumorigenesis”

- The structure of the review seems a bit controversial. For example, the sections 3 and 5, devoted to the functioning in cancer processed are separated by section 4, devoted to the diagnostic and therapy. However, the drug resistance, which is also related to therapy, was discussed it the section 8, after several sections, devoted to the role of vtRNAs in cellular processes. Such constant alternation make the review a bit difficult to follow. I would recommend moving the therapy-related topics in the end of review.

Author Response

(The authors gave the same response as above.)

Round 2

Reviewer 3 Report

Comments and Suggestions for Authors

The manuscript is well written and clear.

Authors should only make small changes in the text and tables.
1.       Use justification in the text
2.       Use the same line spacing
3.       Reduce the font size used in tables and the size of the tables themselves.